# AcquisitionFocus: Joint Optimization of Acquisition Orientation and Cardiac Volume Reconstruction Using Deep Learning

**DOI:** 10.3390/s24072296

**Published:** 2024-04-04

**Authors:** Christian Weihsbach, Nora Vogt, Ziad Al-Haj Hemidi, Alexander Bigalke, Lasse Hansen, Julien Oster, Mattias P. Heinrich

**Affiliations:** 1Institute of Medical Informatics, University of Lübeck, 23562 Lübeck, Germany; z.alhajhemidi@uni-luebeck.de (Z.A.-H.H.); alexander.bigalke@uni-luebeck.de (A.B.); mattias.heinrich@uni-luebeck.de (M.P.H.); 2IADI U1254, Inserm, Université de Lorraine, 54511 Nancy, France; 3EchoScout GmbH, 23562 Lübeck, Germany; lasse@echoscout.ai; 4CHRU-Nancy, Inserm, Université de Lorraine, CIC 1433, Innovation Technologique, 54000 Nancy, France

**Keywords:** cardiac magnetic resonance imaging, shape reconstruction, view optimization, deep learning

## Abstract

In cardiac cine imaging, acquiring high-quality data is challenging and time-consuming due to the artifacts generated by the heart’s continuous movement. Volumetric, fully isotropic data acquisition with high temporal resolution is, to date, intractable due to MR physics constraints. To assess whole-heart movement under minimal acquisition time, we propose a deep learning model that reconstructs the volumetric shape of multiple cardiac chambers from a limited number of input slices while simultaneously optimizing the slice acquisition orientation for this task. We mimic the current clinical protocols for cardiac imaging and compare the shape reconstruction quality of standard clinical views and optimized views. In our experiments, we show that the jointly trained model achieves accurate high-resolution multi-chamber shape reconstruction with errors of <13 mm HD95 and Dice scores of >80%, indicating its effectiveness in both simulated cardiac cine MRI and clinical cardiac MRI with a wide range of pathological shape variations.

## 1. Introduction

Cardiac magnetic resonance (CMR) imaging typically follows a specific routine. Firstly, a low-resolution scout scan is acquired to localize the heart coarsely. Secondly, the scout scan is examined for manual imaging view-plane placement following dedicated protocol guidelines [1]. The scanner is then adjusted to capture the imaging planes of interest. Lastly, the acquired images are examined by clinical experts or automated post-processing software.

### 1.1. MR Physics Constraints and Timing

Examining images relies on sufficient image contrast, i.e., the signal-to-noise ratio (SNR). The SNR of an acquired image slice is constrained by the physical principle of MR as derived by Macovsci [2]:(1)$$\mathrm{SNR}\propto f\left(\mathrm{Obj}\right)\omega_{o}V_{h}\sqrt{T}$$
where $f\left(\mathrm{Obj}\right)$ is the influence of the examined object, $\omega_{0}$ is the resonant frequency, $V_{h}$ is the voxel volume, and *T* is the acquisition time. Consequently, the SNR is affected by the imaging time and the spatiotemporal resolution of a scan. In CMR, the SNR is negatively impacted by cardiac and respiratory motion artifacts that increase with longer acquisition times [1]. Therefore, the acquisition time *T* acts as a lower and upper bound for the quality of the acquired cardiac images. Various sequences have thus been developed to improve the SNR and reduce the acquisition time. The SNR can be increased by combining images of the same cardiac phase when the acquisition is synchronized over multiple heart cycles [1]. This approach often requires breath-holding strategies that burden the patients [3]. In parallel imaging, the acquisition time is shortened by using multiple receiver coils that are read out in parallel [3,4,5]. From another point of view, *T* is proportional to the number of acquired slices $N_{z}$ and the number of acquired K-space lines $N_{y}$, which can be captured at the rate of the repetition time TR [6]:(2)$$T\propto N_{z}N_{y}\mathrm{TR}$$

Equation (Equation 2) states that acquiring more slices at a higher resolution (more K-space lines) takes longer. This has been addressed with compressed sensing where only a fraction of K-space lines are captured, accelerating the acquisition by a constant factor at the cost of introducing artifacts [7]. Nevertheless, applying these techniques for high temporal resolution cine imaging may be insufficient and remains a challenge [8].

In this study, we will investigate a reduced number of imaging slices $N_{z}$ for faster acquisition without necessarily affecting the in-plane resolution or SNR that could additionally be combined with parallel imaging and/or compressed sensing. This reduction is only applicable under the regard that those sparsely acquired slices are sufficiently descriptive for clinical examination. In the cardiac domain, such a sparse stack of slices is frequently acquired along the heart’s short axis to examine the left-ventricular properties that have been proven to contain valuable information for clinical experts [9]. Descriptive imaging planes are also crucial for automated deep learning techniques, which often achieve impressive results but ultimately rely on the data input.

We hypothesize that computer-assisted techniques can benefit from tailoring the slice selection to the automated post-processing task (see Figure 1). For demonstration, we build upon a recent work that explored the challenging task of reconstructing the full cardiac shape from a set of 2D echo views [10]. For MRI, we constrain the acquisition’s field of view to two sparse slices and learn the optimal slice view orientation for accurate shape reconstruction based on coarse localizer information. The definition and selection of optimal imaging planes [1,9,11] for this task may be different from human intuition, especially when deep learning methods are involved. Despite our study being linked to MRI acquisition and (shape) reconstruction, our method is unrelated to image reconstruction from K-space signals. It operates in the image domain after applying the inverse Fourier transform.

### 1.2. Shape Reconstruction and Imaging Plane Optimization

Volumetric shape reconstruction has been previously explored for various medical imaging modality applications. In ultrasound imaging, there is an interest in reconstructing 3D volumes from 2D slice acquisitions of free-hand sweeps. In [12], this was solved by an LSTM model that combined sequential 2D imaging features with accelerometer parameters. Jokeit et al. [13] demonstrated that 3D bone shapes could be reconstructed from standard planar X-ray radiographs using a CycleGAN network. In a similar work, bone structures were reconstructed from sparse view segmentations using neural shape representations [14]. In the cardiac domain, left ventricle shapes were successfully reconstructed from sparse short-axis and long-axis image stacks using deformable mesh priors [15]. Stojanovsi et al. [10] performed reconstruction of the full cardiac shape from multiple slices. To overcome the lack of paired slice and 3D target data, the authors simulated US intensity images for slices that were extracted from a 3D ground-truth mesh. Their approach uses an efficient variant of the Pix2Vox model presented in [16] and will be considered for performance comparison in Section 2.6.

Optimal imaging planes have been considered in [17], where an orthopedic scanning guide for diseases in 3D ultrasound applications was developed. The method relies on a two-stream classification pipeline to predict the probe movement direction and the presence of the desired target view. In the context of MRI, a target view classification network was proposed to determine the optimal MR image slice for detecting lumbar spinal stenosis [18]. The authors selected the optimal image slice from multiple given slices and evaluated the classification outcome for several network architectures and hyperparameters. Cardiac segmentation of the left ventricle and atrium with joint prediction of standard clinical view planes has been previously explored by Chen et al. [19], who aimed to translate findings from automated segmentations into clinical routine protocols. For optimal valvular heart disease assessment, 14 slice orientations were defined using a cardiac MRI reference scan [20]. Odille et al. [21] reconstructed the left ventricular shape by fitting a b-spline model to slice segmentations obtained from motion-corrected high-resolution intensity data. They compared pre-defined configurations of 3–6 sparse slices to evaluate the impact of view plane choices on the shape reconstruction quality. To the best of our knowledge, none of the previously proposed methods studied the joint optimization of view planes and volumetric reconstruction.

### 1.3. Contribution

While previous studies focused on detecting clinical standard imaging planes [15,18,20], we hypothesize that the slice view orientation should be optimized in a task-driven manner and propose the following contributions:In a challenging target scenario, we reconstruct the full cardiac shape of five structures from only two slices.We study the joint optimization of shape reconstruction and view-plane orientation to derive optimal sparse slice configurations.The optimized slice configurations lead to superior reconstruction quality compared to standard clinical imaging planes, which we demonstrate for synthetic and clinically acquired cardiac MRI data.

## 2. Materials and Methods

Our pipeline mimics the MRI acquisition process (see Figure 1): From a low-resolution scout scan, a coarse anatomical shape is generated by image segmentation. We analyze this coarse segmentation to identify standard clinical view planes and optimize the image plane slicing for cardiac shape reconstruction.

### 2.1. Extraction of Clinical Views

Experts follow a semi-automated routine to determine cardiac view planes [22]: Firstly, the left ventricle is localized in the scout scan, then pseudo-two-chamber (2CH) and four-chamber (4CH) views are extracted. Based on these views, a stack of short-axis (SA) images is retrieved, which is a prerequisite to acquiring accurate 2CH and 4CH views. We extract the mentioned views from the coarse image segmentation by analyzing the inertial moments J of the cardiac chamber shapes to construct orthonormal bases for an affine reorientation matrix P,
(3)$$\begin{array}{c} J=\left[\begin{array}{ccc} J_{11} & J_{12} & J_{13}\\ J_{12} & J_{22} & J_{23}\\ J_{13} & J_{23} & J_{33} \end{array}\right]\;J_{ii}=\int_{m}\left(x_{j}^{2}+x_{k}^{2}\right)dm\;J_{ij}=-\int_{m}x_{i}x_{j}dm\;i,j\in \left[1,2,3\right] \end{array}$$
where *m* is the shape’s (voxel) mass, ijk are the spatial indices, and *x* is the distance vector from the point mass to a reference point [23]. The resulting imaging planes are visualized in Figure 2.

### 2.2. Slicing View Optimization

As described in Figure 3, we optimize for affine matrices M that maximize the reconstruction accuracy. We first generate *N* affine matrices M to define the slicing orientation. This work explores the extreme scenario of studying only N=2 slice locations. Subsequently, we apply a reconstruction model to process the extracted slices. The deep learning architecture is laid out more specifically in Figure 4. To obtain optimizable slice orientations, we feed the segmentation of a (low-resolution) scout image scan $V_{in}$ into an acquisition model $A_{i}$. The model comprises two operators: $O_{i}$ aligns the input optimally to yield the oriented volume $V_{or}$. From this volume, the operator *C* extracts a 2D slice *S* per matrix M: (4)$$\begin{array}{cc} O_{i} & :\left\{V_{in}:\Omega_{3D}\to R\right\}\to \left\{V_{or}:\Omega_{3D}\to R\right\},\;i=1,\ldots ,N \end{array}$$(5)$$\begin{array}{cc} C & :\left\{V_{or}:\Omega_{3D}\to R\right\}\to \left\{S:\Omega_{2D}\to R\right\} \end{array}$$

The formulation of $O_{i}$ is inspired by Jaderberg et al. [24] and uses a spatial transformer network to sample an oriented 2D plane from a 3D volume. The network consists of a CNN localization network with learnable parameters $\theta_{O_{i}}$ that maps the input volume $V_{in}$ to six rotational parameters ${\mathrm{ap}}_{i}={\left(ap_{i1},\ldots ,ap_{i6}\right)}^{T}$ and three translational parameters ${\mathrm{tp}}_{i}$ with $3\times N_{tp}$ parameters, where $N_{tp}$ is chosen relative to the target offset space (see Section 2.7). From ${\mathrm{ap}}_{i}$, the rotational components of a 3D affine matrix $M_{i}$ are generated using the continual representation from [25]. The translational vector $t_{i}={\left(t_{i1},t_{i2},t_{i3}\right)}^{T}$ is formulated as:(6)$$t_{\mathrm{ij}}=\frac{2.0}{N_{tp}}\langle softmax\left({\mathrm{tp}}_{\mathrm{ij}}\right),\left(0,1,\ldots ,N_{tp}\right)\rangle -1.0,\;{\mathrm{tp}}_{\mathrm{ij}}\in R^{N_{tp}},j\in \left[1,2,3\right]$$

The 3D affine matrix $M_{i}$ is then used to create a grid for the differentiable spatial transformer sampling layer. A slicing operator, *C*, extracts the center slice of the aligned volume. We want to stress that for every 3D input shape volume, a separate set of ${\mathrm{ap}}_{i}$ is predicted. This enables us to take any segmented input volume and find the correct slicing orientation for the subsequent scans using the same pre-trained model.

### 2.3. Reconstruction Model

For a given set of *N* optimized 2D image slices *S* from the acquisition model, we aim to reconstruct the full volumetric cardiac shape $V_{re}$:(7)$$R:{\left\{S:\Omega_{2D}\to R\right\}}^{N}\to \left\{V_{re}:\Omega_{3D}\to R\right\}$$

Aiming for a mapping $\Omega_{2D}\mapsto \Omega_{3D}$, we configure the model to contain a 2D encoder and a 3D branch, where the inverse of $M_{i}$ is used at the skip connections and the bottleneck to re-embed the 2D slices in 3D space (see Figure 4 and Section 2.7).

### 2.4. Joint Optimization

Given the above models, we obtain *N* optimized slices, by jointly training the parameters of *N* acquisition models $\theta_{O_{1},\ldots ,N}$ and one reconstruction model $\psi_{R}$: (8)$$\begin{array}{cc} V_{or_{1}},\ldots ,V_{or_{N}} & =O_{1}\left(V_{in},\theta_{O_{1}}\right),\ldots ,O_{N}\left(V_{in},\theta_{O_{N}}\right) \end{array}$$(9)$$\begin{array}{cc} S_{1},\ldots ,S_{N} & =C\left(V_{or_{1}}\right),\ldots ,C\left(V_{or_{N}}\right) \end{array}$$(10)$$\begin{array}{cc} V_{re} & =R\left(S_{1},\ldots ,S_{N},\psi_{R_{i}}\right) \end{array}$$

In a simplified setup, where $V_{re}$ and $V_{in}$ have the same spatial resolution, we would require $V_{re}\equiv V_{in}$ for an optimal reconstruction. This mapping could be fulfilled by learning an identity function but is restricted since we feed the data through two bottlenecks that are reducing information by extracting a sparse slice and compressing the shape representation:(11)$$L\left(\theta_{O_{1,\ldots ,N}},\psi_{R}\right)=\ell \left(R_{\psi }\circ C\circ O_{\theta ,1}\left(V_{in}\right),\ldots ,R_{\psi }\circ C\circ O_{\theta ,N}\left(V_{in}\right),V_{re}\equiv V_{in}\right)$$

In our pipeline, the slice bottleneck is particularly interesting, as the reoriented slices $S_{1,\;\ldots ,\;N}$ reveal information about the importance of individual structures for the reconstruction. In an application-oriented setting, the scout scan $V_{in}$ has a lower spatial resolution than the output $V_{re}$. When passing the predicted affine matrix $M_{i}$ to the MRI control panel, the optimized view can be captured in higher resolution to provide more detailed information for the reconstruction (see Figure 3).

### 2.5. Datasets

We performed initial experiments with synthetic cardiac MRI scans generated with XCAT [26] and MRXCAT 2.0 [27]. In this dataset with free-breathing protocol, each scan consists of 100 image frames with 1 mm spatial and 50 ms temporal resolution. The XCAT software provided ground-truth anatomical label maps, whereas texturized MRI simulations were derived from these maps using MRXCAT 2.0. The data were split into 24 training (male phantom) and 16 testing samples (female phantom). To show the effectiveness of our method, a percentage of $\left[25\%\ldots 75\%\right]$ of cardiac phase frames was excluded from the training set to reserve frames of the systolic phase for testing. In subsequent experiments, we used the MMWHS dataset [28] containing 20 labeled, static, nearly isotropic MRI volumes with the following structures: myocardium (MYO), left ventricle (LV), right ventricle (RV), left atrium (LA), and right atrium (RA). The dataset contains significant shape variations, including patients with cardiovascular diseases such as “cardiac function insufficiency, cardiac edema, hypertension […] arrhythmia, atrial flutter, atrial fibrillation, artery plaque, coronary atherosclerosis, aortic aneurysm, right ventricle hypertrophy [, and] dilated cardiomyopathy” [28]. The data were split into training and test data using 3-fold cross-validation.

### 2.6. Experimental Setup and Evaluation

Firstly, in Experiment I, we performed full cardiac shape reconstruction and compared the performance of our model to Pix2Vox (P2V, [16]) and a leaner variant Efficient Pix2Vox (EP2V, [10]), specifically designed for cardiac-slice-to-volume reconstruction (see Section 1.2). In this experiment, we simplified the multi-chamber reconstruction task to a binary shape reconstruction task to match the experimental setup of [10].

Secondly, in Experiment II, we extended the reconstruction task to multiple chambers and investigated the impact of simultaneous view-plane optimization on the reconstruction performance. We conducted an extensive ablation study transitioning from elementary to more elaborate scenarios. This transition involved replacing ground-truth annotations with automated segmentations as well as replacing high-resolution scout scans (1.5 × 1.5 × 1.5 m$m^{3}$/vox) with lower-resolution scout scans (6.0 × 6.0 × 6.0 m$m^{3}$/vox)—a very coarse setting compared to the settings used in [29]. Note that these high-resolution scout scans are not available in clinical settings. Shape reconstruction was performed with just two high-resolution 2D views with 1.5 × 1.5 m$m^{2}$/vox in all scenarios, which can be acquired quickly and enables analysis with high temporal resolution.

Standard clinical views, such as 2CH and 4CH views (see Figure 2) were extracted from the scout input using the method described in Section 2.1. For the MMWHS dataset, we employed 3-fold cross-validation to address significant shape variations in the dataset. We assessed the reconstruction performance with the 95th percentile of the Hausdorff distance (HD95) and Dice score metrics.

### 2.7. Implementation Details

Our acquisition model is a convolutional neural network (CNN) consisting of layers with instance normalization, average pooling, and a final fully connected layer. The last layer maps the input features to six ${\mathrm{ap}}_{i}$ and 3 × $N_{tp}$ values. The affine matrices $M_{i}$ are then constructed using the continual representation of [25] for rotational components and Equation (Equation 6) for translational components, restricting translational shifts to ±20%. The parameter count $N_{tp}=51$ was chosen to be 40% of the spatial input volume length. In preliminary experiments, we attempted to predict the three translational components for every slice with three parameters but experienced instabilities. Mapping the parameters described in Equation (Equation 6) resulted in stable training and improved scores.

The one-hot encoded slice shape output is concatenated channel-wise (see Figure 4, center) and then fed to the reconstruction network. The reconstruction model is a U-Net based on [30], which we configure to consist of a 2D encoder and a 3D decoder by replacing the convolution and normalization layers while keeping the exact kernel sizes. To prevent the U-Net model from sharing information across slices in the encoder, we used grouped convolutions with independent groups per input slice.

The 2D features were re-embedded to the 3D space using the a grid-sampling operator with the inverse affine matrices $M_{i}^{-1}$ for every slice to enable the concatenation of 2D and 3D features at the skip connections. Every block of the reconstruction model (see Figure 4) comprises two (transpose) convolutional operations, followed by instance normalization and LeakyReLU nonlinearities. During joint training, we used the AdamW optimizer [31] $\left(\eta =0.001,\beta_{1}=0.9,\beta_{2}=0.999,decay=0.01\right)$ for the reconstruction model and a batch size of B=4. The acquisition models were optimized using AdamW (η=0.002,decay=0.1) and cosine annealing scheduling with warm restarts [32]. As a loss function, we employed a combination of Dice loss and cross-entropy [30]. We found that simultaneously optimizing both slices resulted in unstable training and, therefore, followed a two-stage approach. First, the slice output of the acquisition model $S_{1}=C\left(O_{1}\left(V_{in}\right)\right)$ was duplicated and stacked across the channel dimension while optimizing the parameters of the CNN. Then, the parameters of model $O_{1}\left(\;\cdot \;\right)$ were fixed, and only the parameters of $O_{2}\left(\;\cdot \;\right)$ were optimized. In both stages, the models were trained for 80 epochs. We always performed a final reconstruction network training from scratch, where the models $O_{1}$, $O_{2}$, and thus the input slices $S_{1}$, $S_{2}$ were fixed. Rotation and scaling augmentation were applied to the input and output shapes to reduce the overfitting of the reconstruction model. For image segmentation, we utilize the U-Net model pipeline of [30], trained on 2D image slices with downsampling augmentation to ensure accurate segmentations for low-resolution and high-resolution inputs.

## 3. Results

### 3.1. Experiment I

The evaluation of reconstruction model performance on the full cardiac shape is shown in Table 1 for the synthetic cine data and in Table 2 for the clinically acquired data. We observed lower Dice scores and higher HD95 errors for the MMWHS dataset, which contains largely varying pathological deformed shapes. Applied to the MRXCAT dataset, our model achieved the lowest HD95 errors in all scenarios and the best Dice score for the p2CH and p4CH slice view inputs. It thus outperformed P2V and EP2V in four of six scores. The P2V model [16] reached the best Dice score when reconstructing MRXCAT data from 2CH and SA views, whereas its efficient variant, EP2V [10], reached the best Dice value on 2CH and 4CH views (see Table 1). When applied to the MMWHS data, our model reached the highest performance in five of six scores, and was only outperformed by EP2V, which presented a lower HD95 error in the case of 2CH and SA view inputs (see Table 2).

### 3.2. Experiment II

We report the results of an extensive ablation study for multi-chamber shape reconstruction with our model on the synthetic MRXCAT dataset in Table 3 and the clinical MMWHS dataset in Table 4, respectively. We compared three ablation scenarios for every dataset, indicated by whitespace in the tables. The top group of values represents the first and most elementary scenario in which high-resolution scouts and ground-truth annotations were considered. The highest HD95 errors were observed for reconstructions based on the p2CH and the p4CH views typically extracted at the start of cardiac routine acquisitions (8.5 and 22.5 mm).

The error was reduced to 6.9 and 14.1 mm for true 2CH and 4CH views (Figure 2). Reconstruction from 2CH + SA yielded errors of 7.6 and 16.0 mm. Randomly chosen views resulted in errors of 8.0 and 17.1 mm (RND, mean out of six runs). Optimizing the views reduced HD95 errors to a lowest of 6.2 and 11.9 mm (−0.8 and −2.2 mm compared to true 2CH and 4CH views). An improvement could likewise be observed for the Dice scores, which improved to 86.9 and 82.7% after optimization.

Figure 5 demonstrates that the highest scores were reached after the second stage of optimization (Section 2.7). In the second ablation scenario, reconstruction from realistic low-resolution scouts and ground-truth annotations was examined (see center groups of Table 3 and Table 4). We only considered the best-performing clinical 2CH + 4CH views from the first scenario for further comparison. For MRXCAT, 7.3 mm HD95 error of 2CH + 4CH views was reduced to 7.0 mm (−0.3) with optimization. While the MMWHS dataset demonstrated a comparable error reduction (−0.7 mm), inferior Dice scores were observed. The last scenario added automated segmentation to the pipeline, resulting in the most application-oriented setting. For the MRXCAT data, HD95 errors increased compared to the ground-truth setting of scenario two, resulting in 13.5 mm for 2CH + 4CH clinical views and 9.7 mm for optimized views. This was not reflected by Dice scores, for which 2CH + 4CH clinical views outperformed the optimized views with 81.0% compared to 79.9% respectively. For the MMWHS data, the reconstruction error increased significantly to 51.2 mm for 2CH + 4CH and 42.6 mm for optimized views. We additionally report volumetric segmentation results for the coarse scout scans. Note that for acquiring the scout scans, 32 captured slices instead of one slice are needed at a lower in-plane resolution (14 per x-, y-axis), increasing acquisition time and making it unsuitable for a direct comparison; hence, the values are enclosed in brackets.

The slicing reorientation obtained for the runs of Table 3 and Table 4 (OPT + OPT) is depicted in Figure 6. Notably, the first view was reoriented from the coronal view to an equivalent of the clinical 4CH view in the first 20 epochs, indicating that the 4CH view contains the most information for reconstruction.

Training and inference were performed on a single NVIDIA TITAN RTX 24 GB graphics card. Each stage of optimization took ∼29 min. Inference took 677 ms for the entire pipeline to reconstruct volumes of 128 × 128 × 128 vox from two 128 × 128 pix slices. Each acquisition model contained 2.8 M parameters, the segmentation model contained 20.7 M parameters, and the reconstruction model contained 15.5 M parameters.

## 4. Discussion

We presented a novel approach to enhance the volumetric reconstruction of cardiac structures from sparse slice acquisitions using joint view-plane location and orientation optimization to overcome scan-time limitations for high-resolution 3D shape reconstructions. We tested our approach on a synthetic, dynamic cine dataset (MRXCAT) and a static dataset (MMWHS) that included significant shape variation caused by pathological deformations.

In the binary cardiac shape reconstruction experiment, our reconstruction model outperformed two related methods with lower HD95 error in five of six scenarios and higher Dice performance in four of six scenarios. Improving on the related methods, we then performed multi-chamber reconstruction and joint optimization of the input views. In an extensive ablation study, we showed that the joint optimization of slicing views could consistently reduce HD95 reconstruction errors across all six of the ablation scenarios we performed (MRXCAT: −0.7 mm, −0.3 mm, −3.8 mm, MMWHS: −2.2 mm, and −0.7 mm, −8.6 mm), whereas two scenarios demonstrated a drop in Dice scores.

For the MRXCAT dataset, a promising low error rate of 9.7 mm HD95 was achieved for multi-chamber reconstruction after view optimization, despite the fact that only a subset of cardiac phases was seen during optimization. This indicates that the reconstruction model learns a generalized shape representation. Visualizing the views of an entire test batch using the heatmap overlay (Figure 6), it is noticeable that views are reoriented consistently to yield optimal reconstruction properties (also refer to Figure 5). For the MMWHS dataset, slice optimization reduced HD95 errors in all scenarios. A significant performance drop was witnessed when slice segmentation was integrated into the pipeline. Here, the slice view segmentation model limits the capability of reconstructing the 3D shape successfully. Pre-training the segmentation model is challenging, as MMWHS data have a large shape-variability and varying contrasts. Moreover, the segmentation model must generalize to arbitrarily oriented 2D slice views that are not constrained to axial, coronal, and sagittal view planes. Training the segmentation model on a larger dataset using the identified optimized slice orientations and spatiotemporal data will certainly further enhance the model’s robustness.

## 5. Conclusions

We showed that five cardiac structures could be reconstructed with <13 mm HD95 and >80% Dice when reconstructing from only two optimized views regarding ground-truth label map inputs. In future work, we plan to investigate the quantification of possible reconstruction errors to assess the applicability of our method in clinical settings. Moreover, the reconstruction from more than two image planes and the determination of the optimal tradeoff between the reconstruction accuracy and the time needed to acquire the slices remains to be explored. The proposed image plane optimization could furthermore be applied to other target tasks, such as pathology classification. Summarizing our approach, we would like to motivate the medical deep learning community to investigate the integration of (slicing) acquisition parameters into their pipelines to improve computer-assisted analysis further.

## Figures and Tables

**Figure 1 sensors-24-02296-f001:**
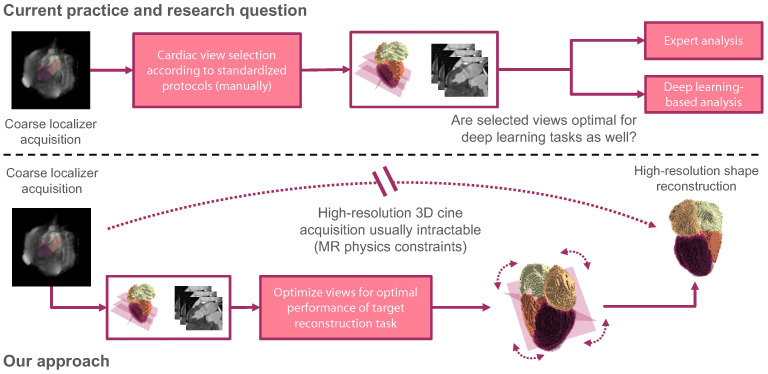
Current practice and research question (**top**): The performance of deep learning-based post-processing methods is restricted by the input data quality, and standardized clinical protocols may be sub-optimal for automated downstream tasks. Our approach and problem setup (**bottom**): Examining cardiac function in high spatial and temporal resolution is desirable, but MR physics constrains the quality of volumetric MR cine acquisitions. We aim to determine optimal descriptive imaging planes for volumetric shape reconstruction from only two view planes.

**Figure 2 sensors-24-02296-f002:**
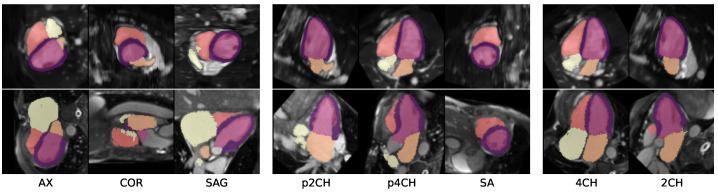
Clinical cardiac views are automatically extracted from the segmentation maps of a coarse scout scan. Axial (AX), coronal (COR), and sagittal (SAG) views are obtained directly from the volume. According to [22], pseudo-two-chamber (p2CH) and four-chamber (p4CH) are then used to plan short-axis (SA) views from which, in turn, accurate 2CH and 4CH views can be retrieved. We mimic this process by analyzing the inertial moments of segmented cardiac chambers.

**Figure 3 sensors-24-02296-f003:**
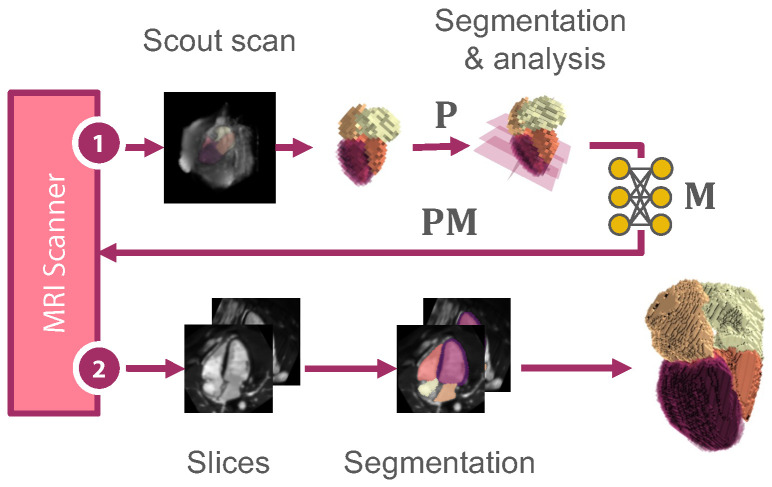
Method overview: From a coarsely segmented scout scan (1), we analyze the cardiac shape, construct affine matrices P representing the standard clinical views, and optimize a neural network to predict a rigid transformation matrix M. This matrix is returned to the scanner to yield optimal slicing parameters for the volumetric shape reconstruction.

**Figure 4 sensors-24-02296-f004:**
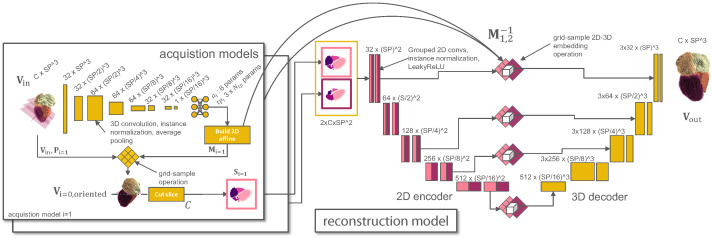
Architecture of the proposed pipeline: The acquisition models (left) optimize the two slicing views (center). The final shape is reconstructed from the stacked slices with a non-symmetric 2D-3D encoder-decoder (right) that contains grouped convolutions in the 2D layers. The 2D-3D skip connections and bottleneck in the reconstruction model are realized using a grid-sample operation that embeds the 2D features in the 3D feature space using the inverse of two affine matrices $M_{1,2}$. (best viewed digitally).

**Figure 5 sensors-24-02296-f005:**
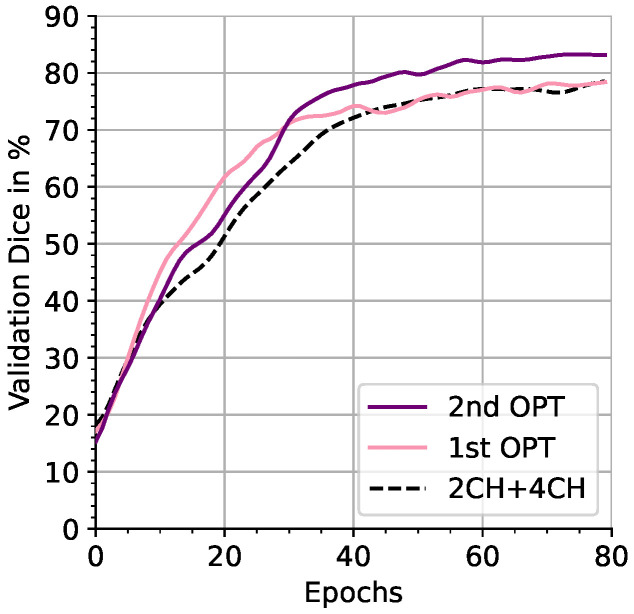
MMWHS Dice scores throughout two-stage training, considering the views 2CH + 4CH as reference. After optimizing the first view, the reconstruction quality is on par with the reference. Optimizing the second view outperforms the reference.

**Figure 6 sensors-24-02296-f006:**
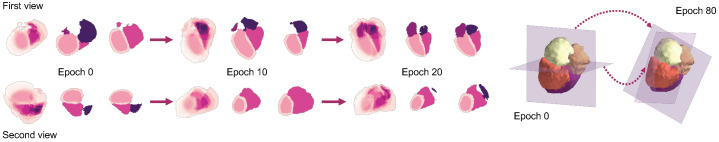
View reorientation during joint training. A heatmap overlay visualizes the orientation across the training batch (left, first column per epoch). Two individual batch samples are displayed in the second and third columns. The first view (**top**) is optimized during the first optimization stage and then fixed in the second optimization stage, in which the second view (**bottom**) is optimized. Notably, the first view was reoriented from the coronal view to an equivalent of the clinical 4CH view in the first 20 epochs. Views are also depicted in 3D, where view planes of epoch 0 were reoriented to view planes of epoch 80, as indicated by the arrows.

**Table 1 sensors-24-02296-t001:** Binary shape reconstruction performance of P2V, EP2V, and our method (see Section 2.3) on the synthetic cardiac data of the MRXCAT dataset.

Synthetic Cine MRXCAT Data			HD95 in mm ↓	Dice in % ↑
**1st View**	**2nd View**	**Model**	μ±σ	μ±σ
p2CH	p4CH	P2V [16]	6.7 ± 2.9	95.4 ± 3.2
		EP2V [10]	7.2 ± 4.6	94.3 ± 4.5
		Ours	**4.7 ± 1.7**	**96.6 ± 1.4**
2CH	4CH	P2V [16]	7.7 ± 5.5	93.6 ± 6.8
		EP2V [10]	5.6 ± 2.4	**96.2 ± 2.1**
		Ours	**5.2 ± 2.8**	95.9 ± 2.2
2CH	SA	P2V [16]	4.6 ± 1.1	**97.1 ± 0.8**
		EP2V [10]	6.2 ± 4.5	95.1 ± 4.8
		Ours	**4.3 ± 2.4**	96.4 ± 2.4

**Table 2 sensors-24-02296-t002:** Binary shape reconstruction performance of P2V, EP2V, and our method (see Section 2.3) on the clinically acquired cardiac data of the MMWHS dataset.

Clinically acq. MMWHS Data			HD95 in mm ↓	Dice in % ↑
**1st View**	**2nd View**	**Model**	μ±σ	μ±σ
p2CH	p4CH	P2V [16]	20.1 ± 6.2	83.0 ± 5.0
		EP2V [10]	22.1 ± 7.2	80.0 ± 7.8
		Ours	**20.0 ± 6.4**	**86.4 ± 4.1**
2CH	4CH	P2V [16]	21.8 ± 5.9	82.5 ± 4.3
		EP2V [10]	22.1 ± 8.4	81.5 ± 7.2
		Ours	**18.1 ± 6.5**	**87.6 ± 3.5**
2CH	SA	P2V [16]	22.6 ± 7.7	82.6 ± 5.4
		EP2V [10]	**20.8 ± 8.1**	83.3 ± 5.2
		Ours	23.7 ± 6.7	**85.4 ± 4.5**

**Table 3 sensors-24-02296-t003:** Multi-chamber shape reconstruction performances for the synthetic cardiac data of the MRXCAT dataset. The scenario’s difficulty increases from the top to the bottom. Bold values indicate the best values obtained within a scenario group of comparable scout resolution and label map settings (ground-truth (GT) or automated segmentation (SG)). Views are indicated by their names, with RND and OPT indicating random selection (mean out of six runs) and the proposed optimization, respectively.

Synthetic Cine MRXCAT Data			HD95 in mm ↓						Dice in % ↑					
**Type of: Scout—Slices**	**1st View**	**2nd View**	**MYO**	**LV**	**RV**	**LA**	**RA**	μ±σ	**MYO**	**LV**	**RV**	**LA**	**RA**	μ±σ
1.5 mm^3^ GT—1.5 mm^2^ GT	p2CH	p4CH	**6.2**	**5.3**	11.9	5.3	13.9	8.5 ± 14.7	**82.4**	**90.0**	84.2	90.6	83.4	86.1 ± 8.5
1.5 mm^3^ GT —1.5 mm^2^ GT	2CH	4CH	6.5	7.1	8.0	5.1	7.7	6.9 ± 2.0	79.9	86.8	83.5	90.7	85.2	85.2 ± 5.9
1.5 mm^3^ GT—1.5 mm^2^ GT	2CH	SA	6.5	7.2	8.6	6.9	8.7	7.6 ± 2.6	79.3	86.5	83.9	88.6	82.9	84.2 ± 6.2
1.5 mm^3^ GT—1.5 mm^2^ GT	RND	RND	7.2	8.4	9.6	8.0	6.9	8.0 ± 5.4	78.9	86.3	84.9	87.1	88.6	85.2 ± 7.0
1.5 mm^3^ GT—1.5 mm^2^ GT	>OPT<	>OPT<	6.3	6.6	**7.1**	**4.6**	**6.3**	**6.2 ± 2.0**	80.7	87.8	**86.3**	**91.0**	**88.9**	**86.9 ± 5.4**
6.0 mm^3^ GT—1.5 mm^2^ GT	2CH	4CH	**6.3**	7.3	10.3	**5.1**	7.6	7.3 ± 3.0	**79.1**	**86.9**	80.7	**91.3**	86.4	84.9 ± 6.7
6.0 mm^3^ GT—1.5 mm^2^ GT	>OPT<	>OPT<	6.8	**7.2**	**6.8**	6.6	**7.4**	**7.0 ± 1.8**	78.7	85.7	**87.3**	88.7	**87.2**	**85.5 ± 6.0**
6.0 mm^3^ SG—N/A	N/A	N/A	(5.3)	(5.3)	(5.5)	(5.6)	(5.8)	(5.5 ± 0.3)	(79.6)	(91.5)	(90.1)	(85.5)	(86.5)	(86.6 ± 4.2)
6.0 mm^3^ SG—1.5 mm^2^ SG	2CH	4CH	10.3	10.2	31.7	**7.3**	7.7	13.5 ± 17.4	68.6	**82.1**	82.4	**86.0**	85.9	**81.0 ± 8.0**
6.0 mm^3^ SG—1.5 mm^2^ SG	>OPT<	>OPT<	**9.4**	**9.8**	**10.0**	11.7	**7.7**	**9.7 ± 3.0**	**69.9**	81.8	**84.0**	76.4	**87.4**	79.9 ± 8.7

**Table 4 sensors-24-02296-t004:** Multi-chamber shape reconstruction performances for the MRI-acquired cardiac data of the MMWHS dataset. The scenario’s difficulty increases from the top to the bottom. Bold values indicate the best values obtained within a scenario group of comparable scout resolution and label map settings (ground-truth (GT) or automated segmentation (SG)). Views are indicated by their names, with RND and OPT indicating random selection (mean out of six runs) and proposed optimization.

Clinically acquired MMWHS data			HD95 in mm ↓						Dice in % ↑					
**Type of: Scout—Slices**	**1st View**	**2nd View**	**MYO**	**LV**	**RV**	**LA**	**RA**	μ±σ	**MYO**	**LV**	**RV**	**LA**	**RA**	μ±σ
1.5 mm^3^ GT—1.5 mm^2^ GT	p2CH	p4CH	7.7	**8.2**	30.3	27.6	38.7	22.5 ± 25.4	78.7	88.3	69.4	75.7	65.4	75.5 ± 16.2
1.5 mm^3^ GT—1.5 mm^2^ GT	2CH	4CH	**6.8**	8.2	19.5	**8.9**	27.1	14.1 ± 10.2	**81.8**	**88.7**	77.2	**86.5**	74.9	81.8 ± 9.5
1.5 mm^3^ GT—1.5 mm^2^ GT	2CH	SA	7.8	10.2	16.5	13.8	31.6	16.0 ± 10.0	79.9	87.7	77.0	79.7	61.3	77.1 ± 12.1
1.5 mm^3^ GT—1.5 mm^2^ GT	RND	RND	12.0	13.9	18.0	18.1	23.2	17.1 ± 10.0	69.3	82.1	**80.4**	78.0	75.5	77.1 ± 9.2
1.5 mm^3^ GT—1.5 mm^2^ GT	>OPT<	>OPT<	8.6	9.7	**15.1**	13.8	**12.1**	**11.9 ± 3.9**	79.7	87.8	79.8	81.1	**85.0**	**82.7 ± 6.5**
6.0 mm^3^ GT—1.5 mm^2^ GT	2CH	4CH	**7.5**	**8.1**	18.9	**11.0**	22.7	13.6 ± 9.2	**81.0**	**89.4**	78.9	**85.2**	76.4	**82.2 ± 8.6**
6.0 mm^3^ GT—1.5 mm^2^ GT	>OPT<	>OPT<	8.9	10.2	**14.8**	16.2	**14.4**	**12.9 ± 7.2**	77.1	86.1	**81.0**	81.3	**81.1**	81.3 ± 9.3
6.0 mm^3^ SG—N/A	N/A	N/A	(10.8)	(12.8)	(16.3)	(12.8)	(13.0)	(13.2 ± 11.5)	(72.3)	(87.6)	(81.7)	(80.0)	(81.0)	(80.5 ± 9.3)
6.0 mm^3^ SG—1.5 mm^2^ SG	2CH	4CH	**17.1**	**19.1**	51.4	64.8	103.8	51.2 ± 50.7	**56.2**	**71.6**	56.3	35.2	38.8	51.6 ± 25.2
6.0 mm^3^ SG—1.5 mm^2^ SG	>OPT<	>OPT<	35.0	32.7	**39.9**	**53.9**	**51.6**	**42.6 ± 23.4**	43.8	69.0	**56.5**	**39.6**	**61.3**	**54.0 ± 19.6**

## Data Availability

Restrictions apply to the availability of XCAT data presented in the referenced work of Segars et al. [26]. The XCAT phantoms were generated with the permission of the Duke University (4D Extended Cardiac-Torso (XCAT) Phantom Version 2.0), DUKE UNIVERSITY, Durham, NC 27708.

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
