# Peer review of "AcquisitionFocus: Joint Optimization of Acquisition Orientation and Cardiac Volume Reconstruction Using Deep Learning"

_sensors, 2024, doi:10.3390/s24072296_

Round 1

Reviewer 1 Report

Comments and Suggestions for Authors

In this manuscript, the authors present a deep learning method that reconstructs the cardiac volume from a subset of slices, while also optimizing the direction of acquisition. The authors validate the performance of the ML-assisted model using two datasets, while illustrating that it achieves low-error volumetric shape reconstruction. The paper is in general interesting, well-written and the references are appropriate. A minor comment:

In my view, the authors should enhance the “Results” section, by including: (i) a small complexity analysis of the DNN model; (ii) some results related to the training time and the inference latency of the model.

Reviewer 2 Report

Comments and Suggestions for Authors

The manuscript under review proposes a deep learning model that reconstructs the cardiac volume from a limited number of input slices while simultaneously optimizing the slice acquisition orientation for this task. Overall, the method and result sections are sound. However, there are several major issues that must be addressed before the work can be published.

Here are my major concerns:

1. In the Introduction section, lines 23, 26-27, 30-32, there are some descriptions about image contrast. However, these descriptions about image contrast are not accurate. So they must be improved. I think the authors need to understand the Magic Triangle in MRI - Time, Resolution and SNR more to write the Introduction about MRI better.

2. For lines 46-55, the authors talked about the acquisition about diffusion MRI, as well as k-space and etc.. However, these are not related to the topic presented in the work.

3. In this work, the authors should clearly state that their work is not in the k-space. It works only in the image space. Otherwise, it is really misleading, especially after reading the title. 

4. I think the whole Introduction needs to be rewritten to motivate the whole work much better. Currently there is so much irrelevant information been included.

Comments on the Quality of English Language

Besides these major concerns, the authors need to proofread the manuscript carefully. Currently there are some typos in the manuscript, such as in line 87, "focussed" should be changed to "focused".

Reviewer 3 Report

Comments and Suggestions for Authors

  1. How does the deep learning model account for variations in cardiac shapes and movements if the training data is not fully representative of all possible pathological variations?
  2. What measures are in place to ensure the model's performance on unseen data, especially when imaging conditions differ significantly from those encountered during training?
  3. How does the model handle poor image quality, including artifacts and noise, in the input slices? Can such issues degrade the quality of the reconstructed cardiac volume?
  4. Is there a clear understanding of the decision-making process of the deep learning model, particularly regarding why certain predictions or reconstructions are made? How does the lack of interpretability affect its clinical utility?
  5. What computational resources are required for training and deploying the deep learning model in clinical settings, and how might these requirements impact its scalability and accessibility?
  6. What strategies are implemented to mitigate the risk of overfitting in the deep learning model, given its susceptibility to memorizing the training data rather than generalizing to unseen data?
  7. How are ethical and legal considerations, such as patient privacy, liability, and regulatory compliance, addressed when deploying the deep learning model in clinical practice?
  8. How adaptable is the model to changes in imaging protocols or hardware, and what challenges might arise from such adaptations in terms of retraining or adjustment?
  9. What steps are taken to clinically validate the effectiveness and safety of the model in real-world patient care scenarios, beyond the promising results observed in simulated and clinically acquired MRI data?
  10. Can the proposed deep learning approach be generalized to other medical imaging modalities or anatomical regions, and what adaptations or validations would be necessary for such generalization?
  11. Why wasn't a conclusion section included in the paper? It's important to add one to effectively summarize and highlight the significance of the research findings.

Round 2

Reviewer 2 Report

Comments and Suggestions for Authors

The authors addressed my concerns for the first round of review. However, after the reading of the new introduction and the revised manuscript, I think there are still some pieces which are missing.

1. Literatures review on the topic. The authors should give references which has been published previously on the same topic.

2. Comparison with the state-of-the-art methods is missing. The authors only present their own results without any comparison with other methods. This makes it hard to evaluate the merit of the current work.

So I think the authors need to include the comparison part to make the current work more complete.

Reviewer 3 Report

Comments and Suggestions for Authors

No comment

Author Response

We want to thank the reviewer again for reviewing our manuscript and for the insightful questions and suggestions for improving our work.

Round 3

Reviewer 2 Report

Comments and Suggestions for Authors

The authors have addressed my concerns and I think it can be accepted for publication.